# Towards a Functional Neuromarker of Impulsivity: Feedback-Related Brain Potential during Risky Decision-Making Associated with Self-Reported Impulsivity in a Non-Clinical Sample

**DOI:** 10.3390/brainsci11060671

**Published:** 2021-05-21

**Authors:** Juliana Teti Mayer, Charline Compagne, Magali Nicolier, Yohan Grandperrin, Thibault Chabin, Julie Giustiniani, Emmanuel Haffen, Djamila Bennabi, Damien Gabriel

**Affiliations:** 1Laboratoire de Recherches Intégratives en Neurosciences et Psychologie Cognitive, Université Bourgogne Franche-Comté, 25000 Besançon, France; charline.compagne@edu.univ-fcomte.fr (C.C.); mnicolier@chu-besancon.fr (M.N.); ygrandperrin@chu-besancon.fr (Y.G.); chabinthibault@gmail.com (T.C.); jgiustiniani@chu-besancon.fr (J.G.); emmanuel.haffen@univ-fcomte.fr (E.H.); dbennabi@chu-besancon.fr (D.B.); dgabriel@chu-besancon.fr (D.G.); 2Service de Psychiatrie de l’Adulte, Centre Hospitalier Universitaire de Besançon, CEDEX, 25030 Besançon, France; 3Centre d’Investigation Clinique, INSERM CIC 1431, Centre Hospitalier Universitaire de Besançon, CEDEX, 25030 Besançon, France; 4Centre Expert Dépression Résistante FondaMental, Centre Hospitalier Universitaire de Besançon, CEDEX, 25030 Besançon, France

**Keywords:** risk-taking, impulsivity, decision-making, electroencephalography (EEG), event-related potentials (ERPs), feedback processing

## Abstract

Risk-taking is part of the multidimensional nature of impulsivity, consisting of an active engagement in behaviors or choices with potentially undesirable results, with probability as the cost for an expected reward. In order to understand the neurophysiological activity during risky behavior and its relationship with other dimensions of impulsivity, we have acquired event-related-potential (ERP) data and self-reported impulsivity scores from 17 non-clinical volunteers. They underwent high-resolution electroencephalography (HR-EEG) combined with an adapted version of the Balloon Analogue Risk Task (BART), and completed the Barratt Impulsiveness Scale (BIS-10) and the Urgency, Premeditation, Perseverance, Sensation Seeking, Impulsive Behavior Scale (UPPS). The ERP components were sensitive to valence (FRN, P300) and risk/reward magnitude (SPN, RewP). Our main finding evidenced a positive correlation between the amplitude of the P300 component following positive feedback and both the global UPPS score and the (lack of) perseverance UPPS subscale, significant for several adjacent electrodes. This finding might suggest an adaptive form of impulsive behavior, which could be associated to the reduction on the difference of the P300 amplitude following negative and positive feedback. However, further investigation with both larger clinical and non-clinical samples is required.

## 1. Introduction

Impulsivity is associated with rapid, premature, inappropriate, unpremeditated behaviors, that often lead to deleterious consequences. Although a recurrent, pathological manifestation of this type of behavior is closely related to suicide risk and represents an important feature of neuropsychiatric diseases, such as borderline personality disorder, antisocial personality disorder, substance dependence, Parkinson’s disease, or attention deficit/hyperactivity disorder (ADHD) [1,2,3], impulsive choices and actions are also part of daily life in non-pathological forms [4]. According to current perspectives, impulsivity is a multidimensional concept comprising, among others, poor inhibition of irrelevant stimuli, increased discounting of delayed rewards, poor planning ability, lack of perseverance, and tendency for risk-taking [5,6,7]. Risky choices involve probability as the cost for an expected reward [8]. In the context of a risky decision, the probabilities in the case of multiple outcomes (including loss or harm) are known [9,10], and risk-taking is understood as actively engaging in behaviors or choices with potentially undesirable results [11]. Risky decision-making has been differentiated from decision-making under uncertainty or ambiguity, during which the probabilities of the outcomes are unknown to the individual [12,13]. 

Several instruments have been developed in order to evaluate impulsive behavior. The available instruments are generally classified as: (i) self-report measures, (ii) behavioral laboratory measures, or (iii) neuropsychological assessments [1,3]. However, when comparing different assessment tools of impulsiveness, significant correlations are often not observed, suggesting the implication of different neural substrates on the behavioral manifestations of impulsivity and reinforcing the hypothesis of its multidimensional nature [6,7]. The lack of a uniform evaluation and of an instrument that correlates with different dimensions of impulsivity represent an important limitation to advances in research and clinical approach [14]. 

When it comes to risk-taking behavior, a commonly applied behavioral laboratory measure is the Balloon Analogue Risk Task (BART) [15]. The aim of this task is to obtain the most money by virtually inflating a series of balloons, knowing that the balloons may explode at any time. A large number of pumps generates greater reward, but usually increases the probability of explosion of the balloon, i.e., the probability of the balloon bursting increases with each pump. Risky behavior is thus measured through the number of balloon pumps that participants make in order to collect the largest possible reward.

The performance in the BART has been shown to correlate with real-life risk-taking, as well as to cigarette smoking, gambling, and drugs and alcohol use [15,16,17,18,19,20]. This makes this task an interesting tool to combine with other instruments, such as neuropsychological assessments, with the aim to establish a link between risky decision-making and its underlying neural processes. Electroencephalography (EEG) and, more precisely, an event-related potential (ERP) technique seem particularly suitable to explore the dynamic aspects of risk-taking behavior during the BART thanks to their millisecond temporal resolution.

There have been several adaptations of the BART to the recording of ERPs, and most studies focused on the early neural response to the reward processing, especially the feedback-related negativity (FRN) and the P300 components [9]. The FRN is defined as the negative deflection, usually peaking between 250–300 ms following feedback onset, and is often associated in the literature to feedback valence and magnitude. The P300 component, on the other hand, is a positive deflection, generally between 300–600 ms after stimulus onset. There is no consensus in the literature regarding its role in the decision-making process. It is believed that the P300 reflects motivational and decisional processes related to the long-term evaluation of outcomes [21]. Likewise, it has been associated with working memory updating, stimulus categorization, strategic processing, evidence accumulation in perceptual decision-making, and indexation of the evaluation of the task relevance of incoming stimuli [9].

Nevertheless, other ERP components, less explored in the context of risky decision-making, have been pointed out as useful to a better understanding of risk proneness. The stimulus-preceding negativity (SPN) is a slow negative wave that increases as the feedback presentation becomes imminent, corresponding thus to the anticipatory phase of reward processing [22]. It is seen as a valuable tool to evaluate uncertainty sensitivity, and stronger SPN components have been associated to higher uncertainty preceding an outcome [23]. Furthermore, another feedback processing component, reward-related positivity (RewP), is believed to index individual differences in relation to reward sensitivity [24]. A blunted RewP activity has been linked to maladaptive risk-taking behavior, involving both risk aversion and risk proneness [25]. However, the electrophysiological studies of risk taking during the BART have not explored the relationship with other dimensions of impulsive-related behavior.

The present study aimed to investigate the associations between neural responses in the context of risk-taking and other impulsive dimensions from a trait-like perspective (not perceived through laboratory measures). We have therefore conducted an exploratory study and our main hypothesis was that the feedback-related potentials, i.e., the FRN and the P300, during the BART, could reflect the behavioral impulsivity assessed through self-report measures. Moreover, since impulsivity is a multidimensional phenomenon, we hypothesized that a combination of the three different classes of instruments previously mentioned would provide an assessment of impulsive behavior that could incorporate both its neurobiological and social aspects. In this sense, we have adapted the BART to ERP recording as a means to investigate the impact of risk and reward magnitude, as well as feedback valence on risk-taking behavior. ERP components were observed in relation to (i) feedback, both at early (FRN) and at late processing stage (P300), and to (ii) risk/reward magnitude, both before stimulus onset, at the reward anticipation phase (SPN), and following stimulus presentation, at the late processing stage (RewP). In addition, we applied the Barratt Impulsiveness Scale (BIS-10) [26] and the Urgency, Premeditation (lack of), Perseverance (lack of), Sensation Seeking, Impulsive Behavior Scale (UPPS) [27]. As a starting point, we enrolled a non-clinical sample, exploring impulsive behavior without pathological manifestations, which could provide important substrate for further research targeting clinical populations.

## 2. Materials and Methods

### 2.1. Participants

Seventeen healthy volunteers were recruited to participate in the current study through social media advertising and the researchers’ social contacts. They were all over 18 years old and right-handed, according to the handedness questionnaire of Oldfield [28]. Eight women and nine men, with a mean age of 28.47 (SD ± 3.37) years old, composed the final sample. In order to be enrolled in the study, all subjects underwent an evaluation with a trained psychiatrist, being assessed for psychiatric disorders and/or ongoing medical treatments. Additionally, they were also screened with validated tools for symptoms of: depression, through the Montgomery–Åsberg Depression Rating Scale (MADRS) and the Beck Depression Inventory (BDI); anxiety, through the Liebowitz Social Anxiety Scale (LSAS); alcohol abuse, through the Alcohol Use Disorders Identification Test (AUDIT); tobacco consumption, through the Fagerström Test for Nicotine Dependence (FTND); and marijuana abuse, through the Diminuer, Entourage, Trop, Cannabis/Cut, Annoyed, Guilty, Eye-opener-cannabis (DETC/CAGE-cannabis). They had no previous medical history of psychiatric disorders, substance or alcohol abuse, neurological diseases, traumatic brain injury, or stroke, nor were they taking any medications at the time of the study. The sample’s psychiatric scores from both clinician- and self-assessment scales are displayed in Table 1. All participants provided written informed consent prior to enrolment. Before starting the BART, participants were informed they would be paid in relation to their performance, in order to enhance their motivation regarding the task. However, at the end of the experiment, they were all equally paid an amount of 75€ for their participation. The study was conducted in accordance with the Declaration of Helsinki and was approved by the Ethics Committee of the University Hospital of Besançon (General Health Administration–ANSM 2016-A00870-51).

### 2.2. Experimental Task and Measurements

This EEG adaptation of the BART offers the possibility to participants to inflate a series of 80 balloons, divided into 4 blocks, with increasing risk of explosion. Each participant received a set of spoken and written instructions, followed by a few practice trials to familiarize themselves with the task prior to its start. During the task, each pump increases the diameter of the balloon and consequently the amount of the reward. The aim of the task is to achieve the highest score, as in the original version of the BART. In contrast with other EEG adaptations of the BART, in which the number of pumps was decreased to avoid the interference of motor activity, in this version we attempted to preserve the motor feature of the original version of the BART by obliging participants to repeatedly pump the balloons in order to increase their size while they could not burst.

In that respect, the task includes two balloon colors: orange and blue (Figure 1). At the beginning of a trial, the balloon is orange. Orange balloons cannot explode, and the participant’s only possibility is to repeatedly inflate them by pressing the key ‘1’. The balloon will remain orange as its diameter increases with pumps, each pump accumulating one point. However, during the same trial, the orange balloon turns blue at a random point between one and five pumps. The screen then freezes, and a fixation cross appears in the center of the blue balloon during an interval between 1 and 1.2 s before the feedback appears. During this interval, the participant no longer has the possibility to inflate the balloon and is instructed to stare at the fixation cross and limit all motor activity. At the end of the interval, the balloon remains blue and there are two possible outcomes to be displayed: (i) a negative one, where the balloon explodes and the gain accumulated on the trial is lost, leading to the beginning of a new balloon trial (starting with an orange balloon), or (ii) a positive one, where the gain accumulated so far on the trial doubles—meaning the same trial is still not over—and the new current score is shown. In the latter, after the positive feedback, the word ‘Cagnotte’ (cash-out) followed by a question mark appears on the blue balloon. At this point, the participant can either choose to keep pumping the balloon in order to increase the trial gains (by pressing the key ‘1’ the balloon becomes orange and can once again be pumped one to five times until it turns blue) or to cash-out the reward accumulated in the trial, permanently saving their earnings to a virtual bank account (by pressing the key ‘2’). In the case they choose to cash-out their trial gains, the total amount saved so far in the virtual bank is displayed and a new trial begins (with an orange balloon).

The number of blue balloons for each trial was randomized between one and twelve, thus the probability of a blue balloon exploding was defined as *p* = 1/(12 − *n*) with *n* = number of blue balloons. For example, the probability of exploding on the first blue balloon is 1/11, the second blue balloon is 1/10, and so on, until the 11th blue balloon. In order to encourage participants to rather keep pumping than cashing-out their earnings after the balloon turned blue, the reward doubled in the case of positive feedback, while only one point would be earned for each pump when the balloon was orange. Participants had the possibility of taking a small break between balloon blocks (every 20 balloon trials). 

### 2.3. Data Acquisition 

EEG signals were recorded using a 256 channel Geodesic Sensor Net (Electrical Geodesics Inc.; EGI, Eugene, OR). All channels were referenced to the vertex (Cz) and collected with a high impedance amplifier, a Net Amp 300 amplifier (Electrical Geodesics) using Net Station 4.5 software (Electrical Geodesics). Continuous recordings were performed with a high-pass set at 0.1 Hz and a sampling rate of 1000 Hz; all channels were referenced to the vertex (Cz) and impedances were below 50 kΩ. Subjects were instructed to limit body movements, eye blinks, and muscular contractions during the task activity and the reward feedback. 

### 2.4. Data Analysis 

#### 2.4.1. Self-Report Measures

From a trait-like perspective, impulsivity was measured through the BIS-10 [26]. The BIS-10 is a self-rated 34 item questionnaire, composed of three subscales: motor-impulsivity, cognitive-impulsivity, and non-planning-impulsivity. The subscale of motor-impulsivity evaluates actions without thinking, cognitive-impulsivity refers to making quick decisions, and non-planning-impulsivity assesses a lack of forethought (“futuring”). Each item is scored on a 0 to 4 points scale. Higher scores indicate higher levels of impulsivity.

This study also included the UPPS [27], a self-rated 45 item scale, evaluating the following dimensions: urgency, (lack of) premeditation, (lack of) perseverance, and sensation seeking. The urgency subscale assesses the tendency of the individual to feel strong impulsions, especially under negative affect; the (lack of) premeditation subscale evaluates their tendency to consider the consequences of an action before initiating it; the (lack of) perseverance, their capacity to sustain their attention during a task that may be hard or tedious; and the sensation seeking subscale involves two aspects: their tendency to enjoy and seek exciting activities and their openness to try new experiences that may or may not be dangerous. Each item is scored on a base of 4 points. The sum of all subscales (global UPPS score) is equally taken into account. Higher scores also indicate higher levels of impulsivity. 

#### 2.4.2. Behavioral Data Analysis

In our adaptation of the BART program, the risk assessment focused on the mean number of adjusted blue balloons only, because it is at this moment that participants have the possibility to place points in the virtual bank account or choose to keep pumping. Precisely, it corresponds to the average number of blue balloons per trial over all the trials that ended with the decision of cashing-out the accumulated points. The average number of blue balloons was calculated in four blocks of 20 trials in order to check whether the risk incurred at the BART was homogeneous throughout the task.

#### 2.4.3. EEG Data Analysis

EEG data analysis was performed using Cartool Software 3.55 (URL: https://sites.google.com/site/cartoolcommunity/files, accessed on 01 February 2019). Raw EEG data were re-referenced offline to a common average reference, a band pass filter was applied between 1 to 30 Hz and a notch filter was applied at 50 Hz to remove environmental artifacts. Analyses were conducted on the interval around the reward screen for two intervals. This method has been applied by our research team in a previous study [29]. 

The processing of the feedback was analyzed through extraction of epochs of 700 ms (100 ms prior to stimulus onset to 600 ms following stimulus onset) from the raw data, with a baseline correction of 100 ms applied before the feedback (100 ms to 0 ms) [29]. Based on grand averages of ERP responses for “doubling” and “explosions” conditions, the FRN was defined as the mean voltage from 270 ms to 345 ms, and the P300 was defined as the mean voltage between 375 ms to 575 ms. The number of accepted trials during the processing of rewards was 22.88 (SD ± 5.79) epochs after a loss and 194.35 (SD ± 47.39) epochs after a gain.

To analyze the influence of the amplitude of the reward, only trials corresponding to the first doubling of the blue balloon and the last doubling of the last blue balloon (i.e., the balloon after which the participant decides to place the points in the bank) were taken into account. The first doubling was thus associated with a small reward and the last doubling with a large reward. Two time intervals were extracted. First, in the anticipation of the reward, epochs of 1000 ms (800 ms prior to the outcome to 200 ms after) were extracted from the raw data, with a baseline correction of 200 ms applied from 1000 ms to 800 ms prior to the reward feedback. The SPN was defined as the mean voltage within 200 ms prior to the reward feedback. In the second temporal interval corresponding to the processing of large and small rewards, epochs of 700 ms (from −100 prior to the stimulus to 600 ms after) were analyzed, with a baseline correction applied before the feedback to the onset of the feedback. For the SPN, an average of 41.53 (SD ± 7.44) epochs were accepted for small rewards and an average of 27.47 (SD ± 8.09) epochs for larger rewards. Based on grand averages of ERPs for “small reward” and “large reward” conditions, the reward positivity was calculated as the mean voltage between 200 and 350 ms. For smaller rewards, an average of 48 (SD ± 9.55) epochs were accepted and an average of 28.18 (SD ± 7.44) epochs for larger rewards.

A semi-automatic artifact rejection method was used with a fixed criterion of ±100 µV. Remaining epochs were visually inspected, manually removing those containing blinks, eye movements, or other sources of transient noise from the analysis. Electrodes with an aberrant signal (excessive noise due to malfunctioning or a bad signal during data collection) were interpolated using a three-dimensional spline algorithm (average: 4.67% interpolated electrodes [30]). Based on previous literature on feedback processing, six central electrodes (Fpz, Fz, FCz, Cz, CPz, and Pz) were chosen for analysis [31,32,33,34,35,36,37].

In order to identify the cortical areas responsible for impulsivity, a source localization of ERPs correlated with impulsivity scales was applied [29] using a distributed linear inverse solution based on a Local Auto-Regressive Average (LAURA) model, comprising a solution space of 3005 nodes to estimate the brain regions in response to the different electrocortical map configurations. The current distribution was calculated within the grey matter of the average brain provided by the Montreal Neurological Institute (MNI). To investigate whether the brain regions were correlated with impulsivity, Cartool software offers the opportunity to select a list of Talairach regions and generates a group of solution points (nodes) that fit within each of the identified regions. To allow current density measures (indicating activation strength in µA/mm^3^) to be extracted from the region of interest, the inverse solution was estimated for the group of solution points for each time window of interest.

#### 2.4.4. Statistical Analysis

The behavioral performance on our adaptation of the BART as it pertains to trials was taken into account by analyzing the task in four blocks with 20 trials each with a one-way repeated-measures analysis of variance (ANOVA). The threshold of significance was set to 5% and post hoc analyses were performed using Bonferroni correction. 

The ERPs analyses were performed with two-way repeated measures ANOVAs with the factor electrodes (FPz, Fz, FCz, Cz, CPz, and Pz) and reward valence (positive/negative outcome) for the processing outcome or the magnitude of the reward (small/large) for the analysis of the reward amplitude.

To identify the presence of a link between the different measures of impulsivity (self-reports, behavioral measures, and ERPs), nonparametric Spearman rank-order correlations were employed. To consider multiple comparisons, the threshold of significance was set to 1%. In the study of correlations with ERP data, the threshold was also set to 1% and had to be present on at least two adjacent electrodes to be considered as significant. 

We performed the analysis using Statistica 11.0 for Windows (StatSoft, Inc., Tulsa, OK, USA).

## 3. Results

### 3.1. Behavioral Results

A block effect was detected on the average number of adjusted blue balloons (F(3.48) = 9.51, *p* < 0.00001). The first block’s average was significantly smaller in comparison to the three following blocks (*p* < 0.01 for all; see Figure 2). We have decided to take into account the second, third, and fourth blocks (60 balloon trials) for subsequent analysis, excluding the first 20 balloon trials. Previous literature suggests that there may be a progressive shift from a context of decision-making under uncertainty to decision-making under risk during the BART [13]. In addition, the difference observed between the first and the subsequent blocks could also have been influenced by a learning or practice effect [38]. During the practice trials and the first block, subjects could still be familiarizing themselves with how the probabilities of explosion work in the task. Hence, the behavior displayed on the last three blocks might represent a more consistent measure of risk-taking in our sample, avoiding confusion with a context of decision-making under uncertainty.

Participants’ mean number of adjusted blue balloons (on the last three blocks) was inversely correlated with their global UPPS score (r = −0.50, *p* < 0.05) and especially with the (lack of) premeditation subscale (r = −0.67, *p* < 0.01) (Figure 3). No other significant correlations were detected between behavioral data and impulsivity scores (Table 1), although a marginally significant correlation between the average number of adjusted blue balloons and the global score of the BIS-10 (r = −0.46, *p* = 0.06) was observed.

### 3.2. ERPs

#### 3.2.1. Feedback-Related Potentials: FRN and P300

A feedback-dependent effect was observed on the FRN component, which displayed larger (more negative) average amplitudes following a negative outcome (balloon explosion) than following a positive outcome (doubling of the score) (F(5.80)= 27.88, *p* < 0.0001; Figure 4). Except for FCz, this difference was present on all other electrodes (*p* < 0.05 for Fz; *p* < 0.01 for Cz and Pz; *p* < 0.001 for FPz and CPz). Correlation analysis did not reveal significant results according to our criterion (*p* < 0.01 on at least two adjacent electrodes, described in the Methods Section).

A feedback-dependent effect was also present on the P300 component (F(5.80) = 8.17, *p* < 0.0001; Figure 4), with larger average amplitudes after negative feedback when compared to positive feedback. This difference between responses was greater on the electrodes Cz (*p* < 0.0001) and CPz (*p* < 0.001). 

The amplitude of the P300 component following positive feedback was negatively correlated with the average number of adjusted blue balloons on the electrodes FPz, Fz, and FCz (FPz: r = −0.49, *p* < 0.05, see Figure 5a; Fz: r = −0.51, *p* < 0.05; FCz: r = −0.6, *p* < 0.05), although this correlation was not significant according to our criterion. On the other hand, the P300 amplitude in response to positive feedback showed a significant positive correlation with the global UPPS score (FPz: r = 0.61, *p* < 0.01; Fz: r = 0.65, *p* < 0.01; FCz: r = 0.67, *p* < 0.01; Cz: r = 0.65, *p* < 0.01) and with its (lack of) perseverance subscale (FPz: r = 0.79, *p* < 0.001, see Figure 5b; Fz: r = 0.76, *p* < 0.001; FCz: r = 0.62, *p* < 0.01). We conducted a power analysis on our main results. For the relationship between the P300 amplitude in response to positive feedback and the global UPPS score, power analysis on FPz = 0.78, on Fz = 0.84, on FCz = 0.87, and on Cz = 0.84. Concerning the relationship between the P300 amplitude in response to positive feedback and (lack of) perseverance subscale, power analysis on FPz = 0.98, Fz = 0.97, and FCz = 0.79. 

In addition, correlations that did not meet our significance criterion were identified (Table 2). The average P300 amplitude observed after a positive outcome on electrodes FPz and Fz was positively correlated with the total score of the BIS and with its cognitive- and motor-impulsivity subscales (*p* < 0.05 for all). The P300 response was also correlated with the (lack of) premeditation subscale of the UPPS, on FPz (*p* < 0.01) and Fz (*p* < 0.05).

#### 3.2.2. Risk/Reward Magnitude-Related Potentials: SPN and RewP

The effect of risk/reward magnitude was observed in the reward anticipation phase, with a more negative SPN when expecting greater (last doubling in the trial) in comparison to smaller (first doubling in the trial) gains (F(5.80) = 3.27, *p* < 0.01). The SPN was significantly more negative on all electrodes (*p* < 0.001 for all; Figure 6a), except for Pz (*p* = 1). No significant correlations with behavioral data or impulsive dimensions were identified. 

Furthermore, differences in the processing of small (first doubling in the trial) compared to greater gains (last doubling in in the trial) were observed on the RewP component (F(5.80) = 4.07, *p* < 0.01). Post hoc analysis confirmed the presence of larger average amplitudes of the RewP following greater gains on Cz (*p* < 0.001; Figure 6b), CPz (*p* < 0.0001), and Pz (*p* < 0.01). No correlations were observed between RewP amplitudes and behavioral or impulsivity measures. 

### 3.3. Source Localization

Given the significant correlations between the P300 response and impulsivity scores, we decided to perform source localization in the time window of the P300 after a reward in order to reveal the cortical areas responsible for this component and potentially involved in impulsive behavior. An activation of the left inferior frontal gyrus and the right parietal area were observed (Figure 7). Correlation analyses performed between the activity of the left inferior frontal gyrus and the right parietal area and impulsivity scales did not reach significance (*p* > 0.1 for all).

## 4. Discussion

This study aimed to explore specific ERP activity related to risk-taking behavior in an adapted version of the BART and possible correlations with other impulsive dimensions in a non-clinical sample. We observed that risk-taking behavior in the task (mean number of adjusted blue balloons) was inversely correlated with the UPPS global and (lack of) premeditation scores (Figure 2). The P300 amplitude following a positive feedback displayed a positive correlation with the UPPS global and (lack of) perseverance scores (Figure 5b). Additionally, EEG data showed that valence affected both early and late feedback-receipt activity (FRN and P300), while risk level and/or gain magnitude affected the reward anticipation activity (SPN) and the early response to a positive feedback (RewP).

The behavioral data suggested that subjects who scored less on the UPPS and on the lack of premeditation (in other words, the ones who usually are able to consider the consequences of their actions before initiating them) would have a greater average number of adjusted blue balloons, being thus more risk prone in this version of the BART. This result may appear counterintuitive. However, the BART depends on learning and constant strategy adaptation during the task, and a poor deliberative process could result in confusion between risk preference and learning ability in the task [39,40]. In other words, it is undetermined whether a participant chooses the risky option due to a risk proneness or due to their baseline strategy. Additionally, it is worth remembering that although our sample could present an impulsive behavior, it did not present any impulsive-related disorders. Risk proneness in this sample would not necessarily mean that they are real-life risk-takers, but possibly that they could be able to display an adaptive impulsive behavior, in line with the task goal of pursuing a high score, in a context where they could earn a real monetary reward without risking real monetary loss.

Moreover, we observed that the feedback processing components (FRN and P300) were sensitive to feedback valence. The FRN following an explosion (loss) event was significantly larger than following a gain, reinforcing the hypothesis that favorable and unfavorable outcomes promote distinct electrophysiological processing, with greater activation in the case of a negative event. This observation is in accordance with previous studies exploring this component in the context of the BART [41,42,43,44], as well as during other risk-taking tasks [9]. Besides the influence of feedback valence, previous research has shown that error-detection activity might also be sensitive to reward magnitude and detection of unexpected outcomes [9], as well as modulated by peer influence [45], executive function [46], trait anxiety [47], and genetic factors [48]. In contrast, reduced FRN amplitudes following negative feedback have been linked to real-life risk-taking [44] and to impulsive-related disorders, such as alcohol use disorder and borderline personality disorder [49,50], indicating a possible association between impaired error detection and maladaptive impulsive behavior. Hence, our participants apparently displayed what would be proper FRN activity, unrelated to impulsivity dimensions.

Still regarding the feedback processing, we observed greater P300 amplitudes following a loss in comparison to a gain (doubling). A more salient P300 component following negative vs. positive feedback is present in the context of the BART for non-clinical subjects [9,41,44] and for patients with alcohol use disorder [49]. Since the BART involves continuous performance evaluation, the detection of larger P300 amplitudes following loss vs. gain may be in accordance with the context-updating hypothesis of the P300 [51,52,53,54,55,56]. According to this model, the P300 is associated to belief-update, whenever new information needs to be integrated into one’s internal mental model. Considering this hypothesis, it seems that negative stimuli elicit greater signaling, allocating more attentional resources for working-memory update (and strategy adaptation) than positive stimuli. 

An important result of the present study is that subjects who displayed smaller P300 amplitudes following gains scored less on the UPPS scale and on the lack of perseverance subscale, being thus less impulsive and having an increased tendency to stay focused on a hard or tedious task, according to these measures. According to this reasoning, subjects who exhibited larger P300 amplitudes following gains—hence reducing the amplitude difference between their response to a loss and a gain—were more impulsive and lacked perseverance. In addition, it is worth mentioning that the P300 amplitude after positive feedback displayed two correlations not considered significant in accordance with our criterion: (i) a negative correlation with the mean number of adjusted blue balloons (Figure 5a) and (ii) a positive correlation with the UPPS lack of premeditation subscale (see Table 2), which suggests that these subjects could also have less regard for the consequences of their actions. The described ERP results are consistent with our previously discussed behavioral data, which suggested that a greater mean number of adjusted blue balloons was curiously linked to increased premeditation ability (lower impulsivity scores), possibly indicating an adaptive risk-taking behavior. 

To the best of our knowledge, no other study has explored the correlations between the UPPS scale and ERP components in the BART. Nonetheless, previous studies applying the BART had detected an association of a reduced difference between the P300 amplitude following negative and positive feedback with impulsive-related behavior [41,44]. This phenomenon was described in non-clinical samples: risky drivers vs. safe drivers [44] and alcohol-intoxicated subjects vs. controls [41]. However, similar results for the P300 were not observed in clinical subjects when it came to binge drinking [57] and alcohol use disorder [49], in comparison to controls.

Considering the aforementioned observations, it is possible to hypothesize that changes on feedback-related responses could index impulsive-related behaviors. A reduction on the amplitude difference between the P300 following negative and positive feedback could be linked to a spectrum of adaptive manifestations of impulsivity. Maladaptive impulsive behavior, on the other hand, could be related to alterations on the FRN amplitude, as suggested by aforementioned studies. However, there is currently insufficient data in order to draw solid conclusions, thus, further research is needed, including larger sample sizes (clinical and non-clinical), which could confirm the significance of a simultaneous correlation of the P300 response with both behavioral and self-report measures.

The localization of brain activity in the time range of the P300 after positive feedback showed an activation of frontal and parietal brain regions, often jointly referred to as the attention network [58]. Both the inferior frontal gyrus and the right parietal cortex are known to be involved in risk evaluation. The structure of the right posterior parietal cortex is associated with decision-making and considered to be a reliable predictor of risky behavior in non-clinical subjects [59]. Additionally, activation of the posterior parietal cortex has been associated with risk preference [12]. The inferior frontal gyrus is a key cortical hub in the circuits of emotional and cognitive control [60]. Although the precise role of the inferior frontal gyrus in decision-making and risk evaluation remains contested, there is more and more support of its implication [61,62]. Previous literature has shown that the activity of the inferior frontal gyrus correlates with risk aversion [63] and more specifically with risk and avoidance (but also with insula and the anterior cingulate cortex) [64,65,66,67]. The insula or inferior frontal gyrus [61], and cingulate regions found to represent variance, have also been proposed to represent alternative risk constructs such as loss probability [68], the magnitude of a loss [64,69], and the related construct of loss aversion [10,70]. Despite these neural structures being involved in risk taking, their activations were correlated neither to behavioral measures at the BART, nor to impulsivity scales. Further explorations with neuroimaging techniques with more accurate spatial resolution such as fMRI may more precisely reveal which cerebral structures are correlated with risk taking in this version of the BART.

We also conducted an exploratory analysis of potentials related to reward and risk magnitude, the SPN and RewP, to evaluate their potential association with impulsivity. During the reward anticipation phase, a significantly larger SPN was observed before greater rewards—which also implied a greater risk of explosion—in comparison to smaller rewards. No relation was found whatsoever between the SPN and impulsive dimensions. To the best of our knowledge, this ERP component had not yet been explored in the context of the BART. Nonetheless, previous studies observing non-clinical subjects during risky decision-making have equally detected increased SPN amplitude in riskier/more uncertain conditions [23,48,71,72]. In the context of our study, a larger SPN may thus suggest that the expected feedback from greater risks and/or with greater rewards (since we are not able to isolate these factors in the present task) may be perceived as more informative and, hence, strongly engage the subjects’ attention for stimulus appraisal and motivational direction of behavior. According to the literature, other factors such as large vs. small presented cues [73], keeping vs. changing the initial choice [74], genetics (Met/Met vs. Val/Val and Val/Met) [48], choice vs. no-choice condition [75], or previous positive vs. negative feedback [76] may also affect the feedback-preceding activity. Impulsive-related clinical populations, on the other hand, have so far showed either no sensitivity to the degree of risk [23,71] or enhanced electrophysiological activity in comparison to controls [77].

The RewP displayed larger amplitudes following greater gains than small gains, being thus affected by reward magnitude and/or risk. Little attention has been given to this component by trials using the BART, but sensitivity to risk and to (low) expectation of gain has also been described, although modulated by age (young adults vs. elderly) in the latter [53]. This component was equally explored in patients suffering from alcohol use disorder vs. controls, but group difference fell short of significance [49]. Finally, in a clinically diverse sample (depression, substance- and alcohol-use disorder), during another uncertain decision-making task, it has been observed that women displayed larger RewP amplitudes than men, and that a blunted response was related to both extremes of high and low risk proneness, even after controlling for psychiatric diagnosis [25]. However, our results did not point towards any relation of the RewP with impulsive dimensions.

Lastly, some limitations and strengths must be taken into consideration. This study is limited by a relatively small sample size, which could compromise generalizability of the findings, especially when it comes to the significant correlations observed. In addition, we lack a model-based approach related to the BART (for examples, see [78,79,80]), and more up-to-date analysis (e.g., whole-brain Multivariate Pattern Analysis [81]) could be applied in association with HR-EEG. However, we chose a classical analytic approach to conduct this exploratory study in order to validate the neural markers of interest specifically in our adaptation of the BART and have the possibility of establishing comparison with previous works in the same field, before reproducing the experiment in a larger trial involving patients suffering from borderline personality disorder [82]. The study’s methodology is nonetheless strengthened by a rigorous selection of non-clinical participants, who displayed varied impulsivity scores, and the application of validated assessment instruments. In addition, when compared to other adaptations of the BART, we believe that the presence of neutral (orange) balloons that must be pressed repeatedly represent an advantage given that it preserves the motor-impulsivity component of the original version of the BART.

Since there is no consensus in the adaptation of the BART to ERP recordings, and that slight changes in the design could modify the impact of learning ability and risk proneness, the presented results might be strictly linked to the design of our adapted version of the BART. To reduce the interference of learning ability during the task, Schonberg et al. [39] suggested separating the learning factor (and thus high epistemic uncertainty) from the BART, for example by giving participants the probabilities of explosion at the beginning of the experiment (see also Pleskac [83]). Another factor that could as well reduce the dependence on the learning process during the experiment is increasing the duration or number of blocks of the training phase. Hence, evaluations of the same task configuration with impulsive pathological populations might be useful to disentangle the underlying mechanisms specifically associated to risk-taking behavior. 

## 5. Conclusions

In this study, we demonstrated a direct relationship between ERP response in the context of risky decision-making and self-rated impulsivity. Our main finding concerned a positive correlation between a late feedback processing component (P300), in the case of positive feedback, and the UPPS global and (lack of) perseverance scores (Figure 5b). Given the potential clinical application of these findings, further research is required towards a better understanding of both adaptive and maladaptive forms of risk-taking and impulsive dimensions, allowing benefits on the approach and follow-up of impulsive behavior. Since a simple combination of different classes of the currently available instruments might not satisfactorily assess impulsivity in its multidimensional nature, the identification of precise dysfunctional mechanisms correlated to impulsive dimensions could contribute to the establishment of neuromarkers of impulsivity, as well as the development of diagnostic and therapeutic tools for impulsive-related disorders. Hence, trials investigating ERP in the context of risk-taking, counting on larger sample sizes, assessing both clinical samples and controls, and applying model-based approaches, should be performed.

## Figures and Tables

**Figure 1 brainsci-11-00671-f001:**
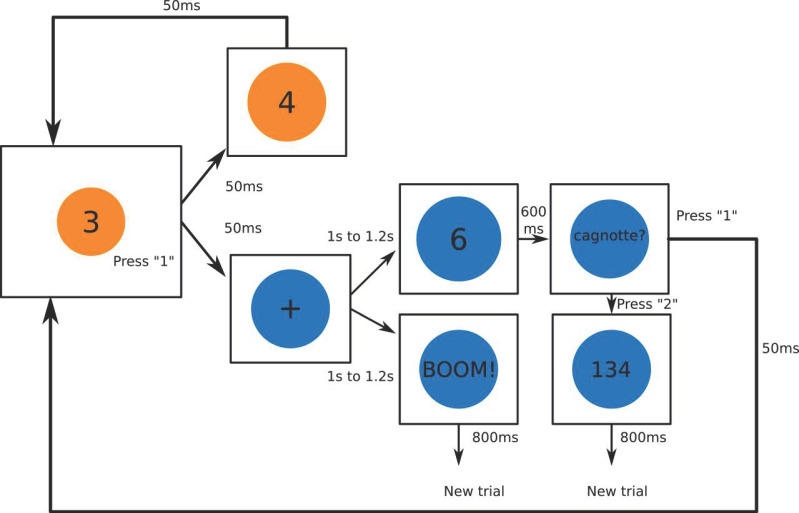
Schematic diagram for the BART adapted to EEG (80 balloon trials split into 4 blocks). At the beginning of each trial, the balloon is orange and cannot explode; the only choice the subjects have is to pump the balloon. Each inflation increases the score in the trial by one point, as displayed in the center of the balloon. At a random point in time, the balloon turns blue, and a fixation cross appears in its center, indicating an imminent outcome. There are two randomly assigned outcome possibilities: (i) balloon explosion—the so far accumulated trial score is lost and a new one automatically starts—or (ii) doubling of the trial score—the new (doubled) score is displayed and in the sequence the word ‘Cagnotte’ (cash-out) followed by a question mark appears on the screen. In the latter, the subjects have the choice to either cash-out the trial gains, ending the current trial and starting a new one, or continue pumping the balloon from where it was (the balloon becomes once again orange), until it turns blue with a fixation cross and a new outcome arrives. During a trial, the probability of a balloon explosion (risk) increases with the number of times the balloon has turned blue.

**Figure 2 brainsci-11-00671-f002:**
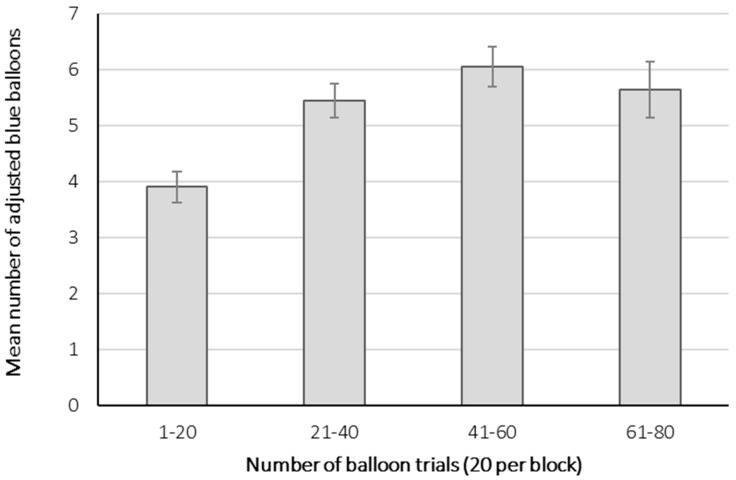
Mean number of adjusted blue balloons per block on the adapted version of the BART. A significant difference was found between the first block and the three consecutive blocks. Error bars represent standard error.

**Figure 3 brainsci-11-00671-f003:**
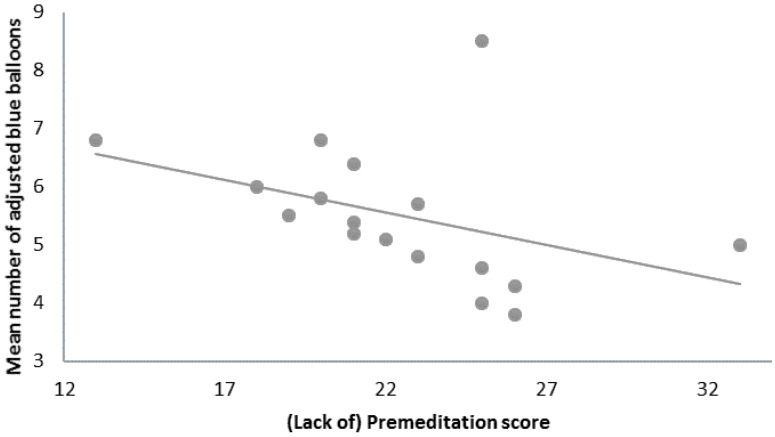
Negative correlation between risk-taking behavior at the BART and the (lack of) premeditation subscale of the UPPS (r = −0.67, *p* < 0.01).

**Figure 4 brainsci-11-00671-f004:**
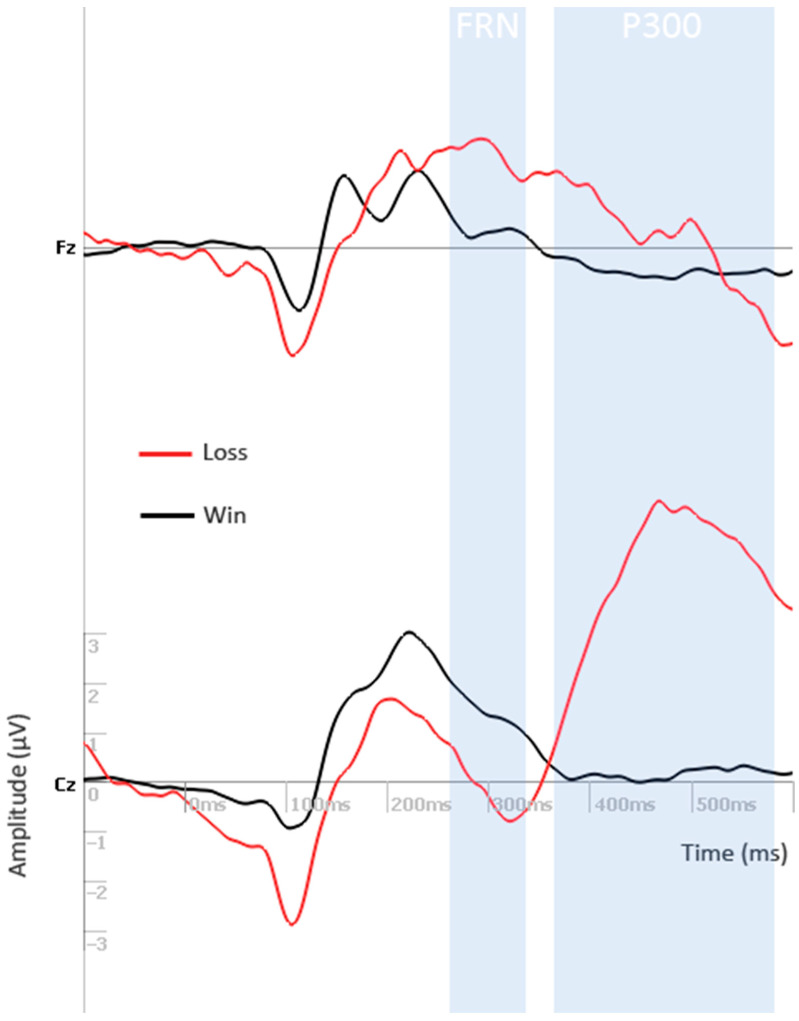
Grand average ERP waveforms after the onset of a loss and a win feedback on electrodes Fz and Cz. The FRN component appears between 275 and 330 ms, while the P300 ERP occurs between 375 and 575 ms after feedback.

**Figure 5 brainsci-11-00671-f005:**
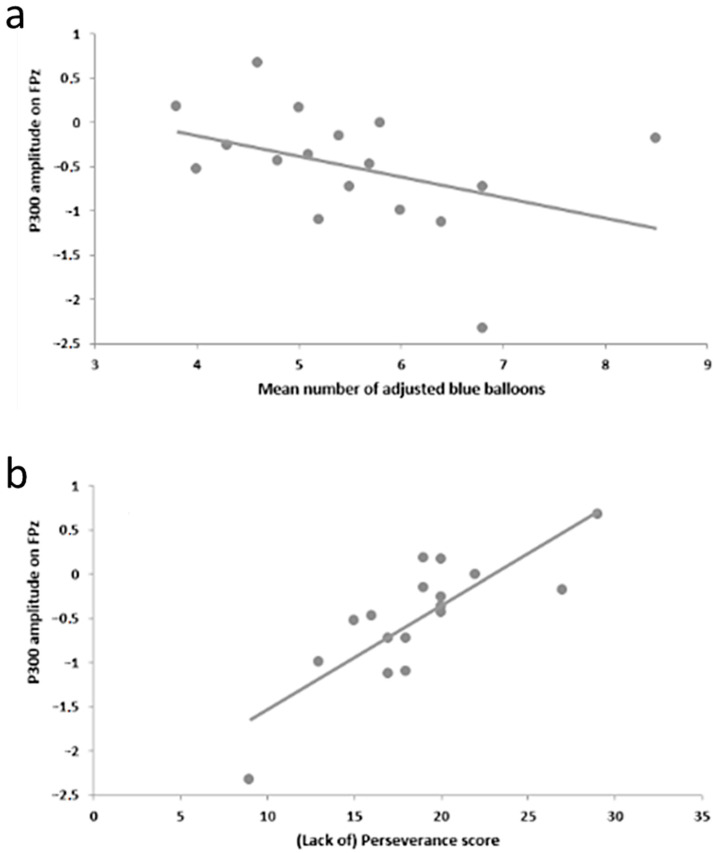
Correlations with the P300 response (μV) after positive feedback on the FPz electrode. (**a**) Negative correlation with the mean number of adjusted blue balloons (r = −0.49, *p* < 0.05). (**b**) Significant positive correlation with the (lack of) perseverance subscale of the UPPS (r = 0.79, *p* < 0.001).

**Figure 6 brainsci-11-00671-f006:**
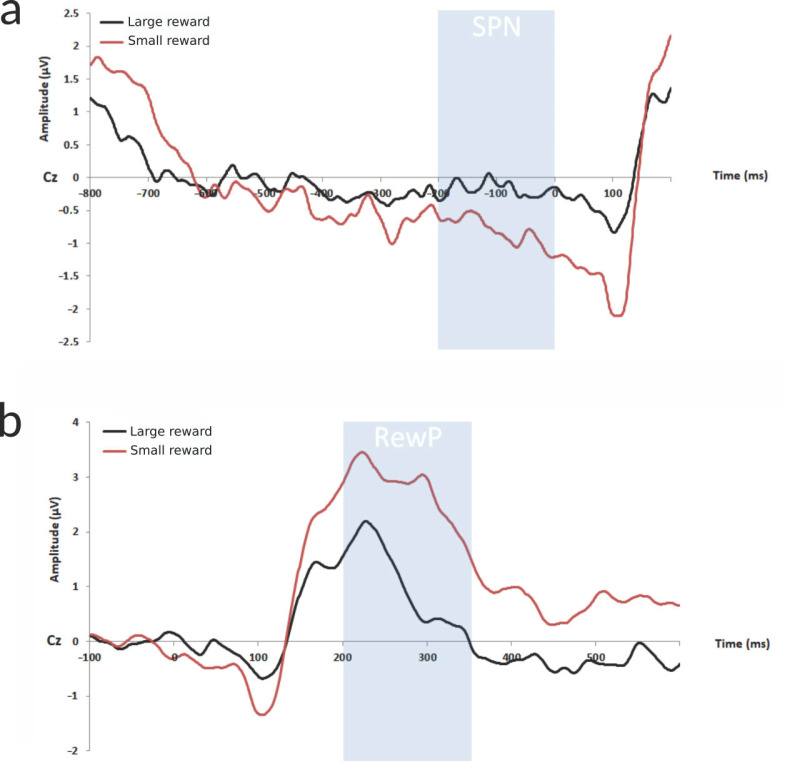
Influence of reward magnitude on the SPN and RewP components on the electrode Cz. (**a**) Neural responses during the anticipation of smaller and larger rewards; a significantly more negative SPN is present for large rewards in the time window of −200 and 0 ms before the feedback appears. (**b**) Neural response during reward processing; a significantly more positive RewP is observed following larger rewards, between 150 and 350 ms after positive feedback.

**Figure 7 brainsci-11-00671-f007:**
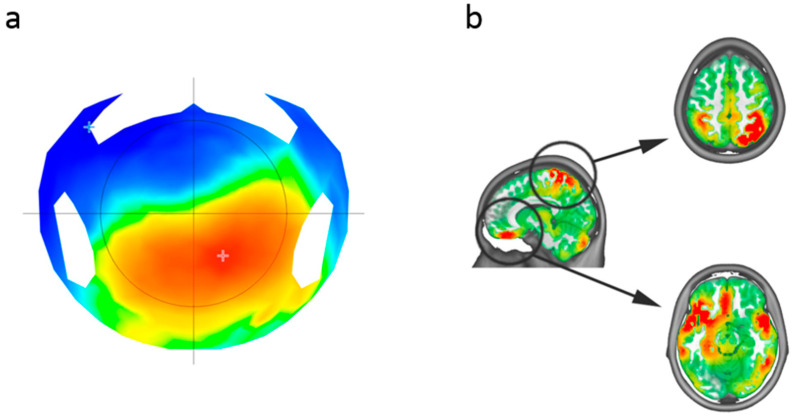
Localization of neural responses relative to a reward in the time window of the P300. (**a**) Topographic representation of neural responses. (**b**) Source localization.

**Table 1 brainsci-11-00671-t001:** Clinician- and self-rated psychiatric scores from participants. MADRS Montgomery–Åsberg Depression Rating Scale; BDI, Beck Depression Inventory; LSAS, Liebowitz Social Anxiety Scale; AUDIT, Alcohol Use Disorders Identification Test; FTND, Fagerström Test for Nicotine Dependence; DETC/CAGE-cannabis, Diminuer, Entourage, Trop, Cannabis/Cut, Annoyed, Guilty, Eye-opener-cannabis; BIS-10, Barratt Impulsiveness Scale; UPPS, Urgency, Premeditation (lack of), Perseverance (lack of), Sensation Seeking Impulsive Behavior Scale.

Measure		Mean	SD	Min	Max
MADRS		2.65	2.91	0	9
BDI		1.70	1.57	0	6
LSAS		27.82	14.67	0	49
AUDIT		3.70	2.34	0	7
FTND (4 smokers)		2.5	1.73	0	4
DETC/CAGE-cannabis		0	0	0	0
BIS-10	Global	50.70	17.49	19	82
	Motor	15.41	5.67	6	28
	Cognitive	18	8.18	5	37
	Non-planning	17.29	7.69	3	30
UPPS	Global	98.76	13.34	73	116
	Urgency	25.88	5.98	18	42
	Premeditation (lack of)	22.41	4.30	13	33
	Perseverance (lack of)	18.76	4.66	9	29
	Sensation seeking	31.70	8.92	17	45

**Table 2 brainsci-11-00671-t002:** Correlations (r-value) between the amplitude of the P300 following positive feedback and self-reported impulsivity scores. UPPS, Urgency, Premeditation (lack of), Perseverance (lack of), Sensation Seeking Impulsive Behavior Scale; BIS-10, Barratt Impulsiveness Scale. * *p* < 0.01, ** *p* < 0.001.

Self-Report Measure		FPz	Fz	FCz	Cz	CPz	Pz
UPPS	Global	0.61 *	0.65 *	0.67 *	0.65 *	-	-
	Urgency	-	-	-	-	-	-
	Premeditation (lack of)	0.63 *	0.58	-	-	-	-
	Perseverance (lack of)	0.79 **	0.76 **	0.62 *	-	-	-
	Sensation seeking	-	-	-	-	-	-
BIS-10	Global	0.5	0.52	-	-	-	-
	Motor	0.58	-	-	-	-	-
	Cognitive	0.54	0.52	-	-	-	-
	Non-planning	-	-	-	-	-	-

## Data Availability

The datasets generated and analyzed during the current study are available in the supplementary file and from the corresponding author on reasonable request.

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
