# Peer review of "Towards a Functional Neuromarker of Impulsivity: Feedback-Related Brain Potential during Risky Decision-Making Associated with Self-Reported Impulsivity in a Non-Clinical Sample"

_brainsci, 2021, doi:10.3390/brainsci11060671_

Round 1

Reviewer 1 Report

In this manuscript, the authors aimed to identify the functional neuromarkers of impulsivity by linking EEG markers to task performance and self-report measures of impulsivity. The strength of this study is using both behavioral task performance and self-report measures. The results of the study may improve our understanding of the neural substrate of impulsivity. However, the manuscript in its current form is poorly written and should be carefully improved. My major concerns are listed below.

Methods: 1), section 2.1, lines 132-134, it seems the authors deceived the participants, was this also approved by the ethics committee of this study?

2), task explanation: lines173-174, the meaning of this sentence is unclear. When the balloon turned blue, how could the participant pump the orange balloon? Furthermore, the image resolution of Fig 1 is too low, it is hard to see the numbers.

3), the self-report measures should be explained in more details, just mentioning the name of the subcomponents is not enough for the readers to understand what it measures. The other scales that reported in Table 1 should also be briefly introduced in the methods section.

Results: 1), the choice of the last three blocks for the calculation of the behavioral marker is too arbitrary. The existence of learning effect does not justify this choice, because behaviors shown during the learning process also reflect impulsivity. The statement "the behavior displaced from the third block would represent a more consistent measure of risk-taking behavior..." is baseless and confusing (the authors did not use the third block alone as an index). I would suggest that the authors also report the correlation results of all the blocks combined (perhaps as supplementary).

2), as my primary major concern, the associations between the behavioral results and the ERPs should be strengthened since these are more directly relevant to the research question of this study, compared to results reported in Figures 4&6. Thus, the ERP indices that showed consistent association with both task performance and self-report measures may be considered more reliable neuromarkers.

3), data reporting: nonsignificant correlations should also be reported (in tables or supplementary).  Figure 2, SD or SE should also be shown. Figure 3&5, coefficient and p values should be shown in the figures. Table 2, colors should be removed and instead please use ** to indicate the significance levels. 

Author Response

Methods:

  • Section 2.1, lines 132-134, it seems the authors deceived the participants, was this also approved by the ethics committee of this study?

“Lines 132-134 Although participation in the study was remunerated by an amount of 75€ at the end of the experiment, participants were informed they would be paid in function of their performance before starting the BART.”

This study was approved by the ethics commitee because all participants received the maximum amount at the end. This is a common approach in this kind of research where it is possible to misinform the patients to encourage them to maximize their performance in order to get the maximum amount of money at the end.

Lines 133-135: “Before starting the BART, participants were informed they would be paid in function of their performance in order to enhance their motivation regarding the task. However, at the end of the experiment, they were all equally paid an amount of 75€ for their participation.”

2) Task explanation: lines 173-174, the meaning of this sentence is unclear. When the balloon turned blue, how could the participant pump the orange balloon? Furthermore, the image resolution of Fig 1 is too low, it is hard to see the numbers.

Thank you for the remark, once the orange balloon turns blue (and if it does not explode) the gain doubles and the participants have the coice to cash-out or to keep on pumping the balloon. If they do so, the color of the balloon becomes orange again to allow pumping until the next blue balloon appears. We have improved the figure quality as well as the task description:

Lines 150-179: “In that respect, the task includes two balloon colors: orange and blue (Figure 1). At the be-ginning of a trial, the balloon is orange. Orange balloons cannot explode and the partici-pant’s only possibility is to repeatedly inflate them by pressing the key ‘1’. The balloon will remain orange as its diameter increases with pumps, each pump accumulating one point. However, during the same trial, the orange balloon turns blue at a random point between one and five pumps. The screen then freezes, and a fixation cross appears in the center of the blue balloon during an interval between 1 and 1.2sec before the feedback ap-pears. During this interval, the participant no longer has the possibility to inflate the bal-loon and is instructed to stare at the fixation cross and limit all motor activity. At the end of the interval, the balloon remains blue and there are two possible outcomes to be dis-played: (i) a negative one, where the balloon explodes and the gain accumulated on the trial is lost, leading to the beginning of a new balloon trial (starting with an orange bal-loon); or (ii) a positive one, where the gain accumulated so far on the trial doubles – meaning the same trial is still not over – and the new current score is shown. In the latter, after the positive feedback, the word ‘Cagnotte’ (cash-out) followed by a question mark appears on the blue balloon. At this point, the participant can either choose to keep pumping the balloon in order to increase the trial gains (by pressing the key ‘1’ the balloon becomes orange and can once again be pumped one to five times until it turns blue), or to cash-out the reward accumulated in the trial, permanently saving their earnings to a vir-tual bank account (by pressing the key ‘2’). In case they choose to cash-out their trial gains, the total amount saved so far on the virtual bank is displayed and a new trial begins (with an orange balloon).

The number of blue balloons for each trial was randomized between one and twelve, thus the probability of blue balloon exploding was defined as p=1/12-n with n= number of blue balloons. For example, the probability of exploding on the first blue balloon is 1/11, the second blue balloon is 1/10, and so on, until the 11th blue balloon. In order to encourage participants to rather keep pumping than cashing-out their earnings after the balloon turned blue, the reward doubled in case of a positive feedback, while only one point would be earned for each pump when the balloon was orange. Participants had the possibility of taking a small break between balloon blocks (every 20 balloon trials).”

  • The self-report measures should be explained in more details, just mentioning the name of the subcomponents is not enough for the readers to understand what it measures. The other scales that reported in Table 1 should also be briefly introduced in the methods section.

Thank your for this observations, we have modified the manuscript accordingly:

Lines 196-203: From a trait-like perspective, impulsivity was measured through the BIS-10 [25]. The BIS-10 is a self-rated 34 item questionnaire, composed by three subscales: motor-impulsivity, cognitive-impulsivity, and non-planning-impulsivity. Each item is scored on a 0 to 4 points scale. Higher scores indicate higher levels of impulsivity. This study also included the UPPS [24], is a self-rated 45 item scale, evaluating the following dimensions: urgency, (lack of) premeditation, (lack of) perseverance, and sensation seeking. Each item is scored on a base of 4 points. The sum of all subscales (global score) is equally taken into account. Higher scores also indicate higher levels of impulsivity.

Lines 127-133: “In order to be enrolled in the study, all subjects underwent an evaluation with a trained psychiatrist, being assessed for psychiatric disorders and/or ongoing medical treatments. Additionally, they were also screened with validated tools for symptoms of: depression, through the Montgomery–Åsberg Depression Rating Scale (MADRS) and the Beck De-pression Inventory (BDI); anxiety, through the Liebowitz Social Anxiety Scale (LSAS); al-cohol abuse, through the Alcohol Use Disorders Identification Test (AUDIT); tobacco con-sumption, through the Fagerström Test for Nicotine Dependence (FTND); and marijuana abuse, through the Diminuer, Entourage, Trop, Cannabis/Cut, Annoyed, Guilty, Eye-opener-cannabis (DETC/CAGE-cannabis; French version).”

Results: 1) The choice of the last three blocks for the calculation of the behavioral marker is too arbitrary. The existence of learning effect does not justify this choice, because behaviors shown during the learning process also reflect impulsivity. The statement "the behavior displaced from the third block would represent a more consistent measure of risk-taking behavior..." is baseless and confusing (the authors did not use the third block alone as an index). I would suggest that the authors also report the correlation results of all the blocks combined (perhaps as supplementary).

Thank you for this remark. We added the correlation results of all block combined as supplementary material. However, we have to mention that this was not an arbitrary choice. Literature suggests that there may be “a gradual shift from decision-making under uncertainty to decision-making under risk” during the BART (https://www.frontiersin.org/articles/10.3389/fpsyg.2018.02194/full). At early stages, the epistemic uncertainty of the individual is high, and pumping and errors might be due to exploration. After collecting a certain amount of trials, the epistemic uncertainty related to the knowledge of the latent parameter regulating observations (the bursting probability) decreases. We precisely wanted to avoid epistemic uncertainty in our measures. We have previously shown that in the first block of the BART, performances are different than in other  blocks, certainly because at this stage behavior mixes uncertainty and risk-taking (https://pubmed.ncbi.nlm.nih.gov/31554273/ ). The fact that there were stronger effects on the last three blocks was rather expected.

We have clarified the matter on the Results section:

Lines 291-300: “We have decided to take into account the second, third, and fourth blocks (60 balloon tri-als) for subsequent analysis, excluding the first 20 balloon trials. The literature suggests that there may be a progressive shift from a context of decision-making under uncertainty to decision-making under risk during the BART. In addition, the difference observed be-tween the first and the subsequent blocks could as well have been influenced by a learning or practice effect [36]. During the practice trials and the first block, subjects could still be familiarizing with how the probabilities of explosion work in the task. Hence, the behavior displayed on the last three blocks might represent a more consistent measure of risk-taking in our sample, avoiding confusion with a context of decision-making under uncertainty.”

2) As my primary major concern, the associations between the behavioral results and the ERPs should be strengthened since these are more directly relevant to the research question of this study, compared to results reported in Figures 4&6. Thus, the ERP indices that showed consistent association with both task performance and self-report measures may be considered more reliable neuromarkers.

Thank you for this valuable remark. We have modified key paragraphs to highlight the potential of our ERP findings as neuromarkers as follows:

First paragraph of the Discussion section:

Lines 402-413: “This study aimed to explore specific ERP activity related to risk-taking behavior in an adapted version of the BART, and possible correlations with other impulsive dimensions in a non-clinical sample. We observed that risk-taking behavior in the task (mean number of adjusted blue balloons) was inversely correlated with the UPPS global and (lack of) premeditation scores (Figure 2 for the latter). The mean number of adjusted blue balloons was also inversely correlated to the amplitude of late feedback processing (P300) in case of a positive feedback. The P300 amplitude following a positive feedback also displayed a positive correlation with the UPPS global and (lack of) perseverance scores (Figure 5 for the latter). Additionally, EEG data showed that valence affected both early and late feed-back-receipt activity (FRN and P300), while risk level and/or gain magnitude affected the reward anticipation activity (SPN) and the early response to a positive feedback (RewP).

Conclusion:

Lines 560-566: “In this study, we demonstrated a direct relationship between ERP response in the context of risky decision-making and self-rated impulsivity. Our main finding concerned a positive correlation between a late feedback processing component (P300), in case of a positive feedback, and the UPPS global and (lack of) perseverance scores (Figure 5). Given the potential clinical application of these findings, further research is required towards a better understanding of both adaptive and maladaptive forms of risk-taking and impul-sive dimensions, allowing benefits on the approach and follow-up of impulsive behavior.”

3) Data reporting: non significant correlations should also be reported (in tables or supplementary). Figure 2, SD or SE should also be shown. Figure 3&5, coefficient and p values should be shown in the figures. Table 2, colors should be removed and instead please use ** to indicate the significance levels.

Thank you for the suggestions, the figures and table have been modified accordingly. The correlation analysis have equally been added as supplementary material.

Reviewer 2 Report

The authors present an exploratory analysis to assess the relationship between personality traits, behavioral measures of risky decision-making, and EEG. I think there is still room for improvement, from both a conceptual and analytic perspective. Some details about analysis and experimental framework have to be explained and justified more clearly. My comments are shown below.

Main

Line 58. You state that “the lack of a uniform assessment […] clinical approach”. However, your hypothesis (e.g. in Line 106) is based on the optimistic view that you are actually conceptualizing a comprehensive assessment, maybe intended to solve the aforementioned limitations. I really think that, in general, even positive results of your approach don’t provide proof that your approach is a “comprehensive approach”. The first reason is that you are making inferences on a model-based perspective, such as Joint Modeling, which really provides a framework for a “comprehensive” approach. The second reason is that you are still adopting self-reports, which are soft behavioral measures. You can choose whether you want to re-postulate, or scale back, or even explain better, your concept of “comprehensive assessment”.

Line 88. Here, you are providing a background of the studies investigating risk-related behavior. However, you named the “context of uncertainty” as pertinent to the understanding of risky behavior. And I agree of course. My problem is that, during the text, I could not see an attempt to disentangle, conceptually, the difference between uncertainty-drive behavior and risky-related behavior. I would say that, in general, avoiding considering such a disentangling, might drive to poor conclusions and poor quality of the treatment of the topic. Behaviors that might seem risky, can be due to the need to explore the environment, before exploiting it. For instance, the individual approaching the task might be uncertain about the true latent parameter in the environment which generates observations. In the BART context, the latent parameter might be the probability of an explosion. At early stages, the epistemic uncertainty of the individual is high, and pumping and errors might be due to exploration. After collecting a certain amount of trials, the epistemic uncertainty related to the knowledge of the latent parameter regulating observations (the bursting probability) decreases. In this case, exploiting the environment, having the knowledge of, say, high bursting probability, might be related to risk. To summarize, when epistemic uncertainty is high, defective behavior might not be due to risk. See “Disentangling Risk and Uncertainty: When Risk-Taking Measures Are Not About Risk” by De Groot and Thurik (2018), for an informal treatment of something related to what I’ve been discussing. You might be interested (and it is not mandatory at all) in including such considerations in your discussion, especially in relation to your results. I’m just curious about that.

Line 121. Can you please add a power analysis since the sample size seems pretty small, especially in relation to the EEG analysis you performed? In my opinion, power analysis results don’t invalidate suitability for publication, but it might be useful to readers intended to use conclusions coming from this study.

Lines 158 – 169. I think the description of the alternatives should be revised a little. The possible flow of the events starts with the possible explosion of the blue balloon, as it first appears if I understood correctly. This should be presented first. After the screen freezes, the fixation cross appears. Which is the interval between the fixation cross and the eventual first explosion? Then, if the blue balloon doesn’t explode, the individual can start pumping. After starting pumping, how is the probability of explosion regulated? In general, I think you should embed the two alternatives description with the previous text in a more logical flow, despite the fact the figure is provided or not.

Line 176. This is just a personal opinion, you might want to ignore. I noticed you used a 256 channels device to record neural data, but then you focus on just 6 of them. I think this is really a waste of information, especially when you will is to propose a “comprehensive” approach that should overcome the limitations of standard ones. In relation to this, you used a really retrograde and outdated analytic framework (e.g. ANOVA). You might be interested in more up-to-date approaches, such as whole-brain (although not necessarily) Multivariate Pattern Analysis.

Line 197. “The average number of blue balloons”. Do you mean the balloons exploded after pumping?

Line 200. Honestly, I didn’t get when do you exactly pre-process EEG data. You did pre-processing on extracted epochs. If this is the case, why not pre-process the whole time series before epoching?

Line 230. Is there any particular reason why you filtered ERPs and not raw data? Maybe you wanted to refer to epochs, and not ERPs. Results change considerably if you decide to filter epochs or ERPs. You should discuss your choices more clearly.

Line 251. Is there any particular reason why you have chosen to perform one-way repeated measures ANOVA? Which is the variance between clusters (individuals). You might be interested in the Intra-Class correlation index, and based on that you should opt for a Mixed-Effects modeling approach.

Line 374. I think it is worth mentioning that the study lacks a model-based approach, which might be further investigated. Nowadays, we have sophisticated psychometric tools to model behavioral data, since extracting a summary measure (e.g. number of pumps) might not represent a valid account of some cognitive construct. One way to solve such a measurability issue is to use cognitive model decomposition to extract cognitive parameters from behavioral data. There are some examples using such an approach for the BART out there, such as, “D’Alessandro et al. (2020) – A joint modeling approach to analyze risky decisions by means of diffusion tensor imaging and behavioral data”; “Van Ravenzwaaij et al. (2011) – Cognitive model decomposition of the BART: assessment and application”, “Rolison et al. (2012) – Risky decision making in younger and older adults: the role of learning.”, to name just a few. The idea is to use cognitive parameters instead of summary measures for further analysis.

Figures

Figure 4 and 6. You should report exactly what the grey areas refer to in the caption.

Typos and Formatting

Is there any particular reason (e.g. editorial) why you have reference indexes in isolated brackets, that is, “[i],[j]”, instead of “[i,j]”.

Why there is no spacing between a word and the reference index bracket, that is, “word[i]”, instead of “word [i]”.

Line 58. “an uniform” should be “a uniform”. In general, the English quality is poor for the whole sentence.

Line 124. Remove “+/-“ above “SD”, here and in other places.

Author Response

Line 58. You state that “the lack of a uniform assessment […] clinical approach”. However, your hypothesis (e.g. in Line 106) is based on the optimistic view that you are actually P 0 WORDS Paragraphconceptualizing a comprehensive assessment, maybe intended to solve the aforementioned limitations. I really think that, in general, even positive results of your approach don’t provide proof that your approach is a “comprehensive approach”. The first reason is that you are making inferences on a model-based perspective, such as Joint Modeling, which really provides a framework for a “comprehensive” approach. The second reason is that you are still adopting self-reports, which are soft behavioral measures. You can choose whether you want to repostulate, or scale back, or even explain better, your concept of “comprehensive assessment”

Thank you for this remark. The term “comprehensive assessment” was indeed not properly chosen, we have decided repostulate the following passages:

Lines 58-60: “The lack of a uniform evaluation and of an assessment tool that correlates with different dimensions of impulsivity represent an important limitation to advances on research and clinical approach [12].”

Lines 105-108: “Moreover, since impulsivity is a multidimensional phenomenon, we hypothesized that a combination of the three different classes of instruments previously mentioned would provide an assessment of impulsive behavior that could incorporate both its neurobiological and social aspects.”

Line 88. Here, you are providing a background of the studies investigating risk-related behavior. However, you named the “context of uncertainty” as pertinent to the understanding of risky behavior. And I agree of course. My problem is that, during the text, I could not see an attempt to disentangle, conceptually, the difference between uncertainty-drive behavior and risky-related behavior. I would say that, in general, avoiding considering such a disentangling, might drive to poor conclusions and poor quality of the treatment of the topic. Behaviors that might seem risky, can be due to the need to explore the environment, before exploiting it. For instance, the individual approaching the task might be uncertain about the true latent parameter in the environment which generates observations. In the BART context, the latent parameter might be the probability of an explosion. At early stages, the epistemic uncertainty of the individual is high, and pumping and errors might be due to exploration. After collecting a certain amount of trials, the epistemic uncertainty related to the knowledge of the latent parameter regulating observations (the bursting probability) decreases. In this case, exploiting the environment, having the knowledge of, say, high bursting probability, might be related to risk. To summarize, when epistemic uncertainty is high, defective behavior might not be due to risk. See “Disentangling Risk and Uncertainty: When Risk-Taking Measures Are Not About Risk” by De Groot and Thurik (2018), for an informal treatment of something related to what I’ve been discussing. You might be interested (and it is not mandatory at all) in including such considerations in your discussion, especially in relation to your results. I’m just curious about that.

Thank you for pointing out this important matter, we fully agree with this comment. We precisely wanted to avoid the part of high epistemic uncertainty in our measures, by removing the first block. We have previously shown that in the first block of the BART, performances are different than in other  blocks, certainly because at this stage behavior mixes uncertainty and risk-taking (https://pubmed.ncbi.nlm.nih.gov/31554273/ ). The following passages have been modified in order to address the difference between decision-making under risk and uncertainty:

Lines 47-53: “Risky choices involve probability as the cost for an expected reward [8]. In the context of a risky decision, the probabilities in case of multiple outcomes (including loss or harm) are known [9, 10], and risk-taking is understood as actively engaging in behaviors or choices with potentially undesirable results [11]. Risky decision-making has been differentiated from decision-making under uncertainty or ambiguity, during which the probabilities of the outcomes are unknown to the individual.”

Lines 294-302: “Previous literature suggests that there may be a progressive shift from a context of decision-making under uncertainty to decision-making under risk during the BART. In addition, the dif-ference observed between the first and the subsequent blocks could as well have been in-fluenced by a learning or practice effect [36]. During the practice trials and the first block, subjects could still be familiarizing with how the probabilities of explosion work in the task. Hence, the behavior displayed on the last three blocks might represent a more con-sistent measure of risk-taking in our sample, avoiding confusion with a context of deci-sion-making under uncertainty.”

Line 121. Can you please add a power analysis since the sample size seems pretty small, especially in relation to the EEG analysis you performed? In my opinion, power analysis results don’t invalidate suitability for publication, but it might be useful to readers intended to use conclusions coming from this study

It is difficult to give a proper power analysis because this was an exploratory study with several aims. This study has to be considered as anciliary from a study on patients where a power study was classicaly conducted. However, we conducted a power analysis from our main results, i.e. the relationship between the P300 amplitude in response to positive feedback and the global UPPS score. Power on FPz=0.78 ; Fz=0.84 ; FCz=0.87 ; Cz=0.84. Concerning the relationship between the relationship between the P300 amplitude in response to positive feedback and (lack of) perseverance subscale, power FPz = 0.98 ; Fz = 0.97 ; FCz = 0.79.

Lines 314-319: “We conducted a power analysis on our main results. For the relationship between the P300 amplitude in response to positive feedback and the global UPPS score, power analy-sis on FPz=0.78, on Fz=0.84, on FCz=0.87, and on Cz=0.84. Concerning the relationship between the P300 amplitude in response to positive feedback and (lack of) perseverance subscale, power analysis on FPz=0.98, Fz=0.97, and FCz=0.79.”

Lines 158 – 169. I think the description of the alternatives should be revised a little. The possible flow of the events starts with the possible explosion of the blue balloon, as it first appears if I understood correctly. This should be presented first. After the screen freezes, the fixation cross appears. Which is the interval between the fixation cross and the eventual first explosion? Then, if the blue balloon doesn’t explode, the individual can start pumping. After starting pumping, how is the probability of explosion regulated? In general, I think you should, despite the fact the figure is provided or not.

Thank you for your suggestions. We have adapted the task description and integrated the two outcomes’ description with the previous text in a more logical flow:

Lines 149-178: “In that respect, the task includes two balloon colors: orange and blue (Figure 1). At the be-ginning of a trial, the balloon is orange. Orange balloons cannot explode and the partici-pant’s only possibility is to repeatedly inflate them by pressing the key ‘1’. The balloon will remain orange as its diameter increases with pumps, each pump accumulating one point. However, during the same trial, the orange balloon turns blue at a random point between one and five pumps. The screen then freezes, and a fixation cross appears in the center of the blue balloon during an interval between 1 and 1.2sec before the feedback ap-pears. During this interval, the participant no longer has the possibility to inflate the bal-loon and is instructed to stare at the fixation cross and limit all motor activity. At the end of the interval, the balloon remains blue and there are two possible outcomes to be dis-played: (i) a negative one, where the balloon explodes and the gain accumulated on the trial is lost, leading to the beginning of a new balloon trial (starting with an orange bal-loon); or (ii) a positive one, where the gain accumulated so far on the trial doubles – meaning the same trial is still not over – and the new current score is shown. In the latter, after the positive feedback, the word ‘Cagnotte’ (cash-out) followed by a question mark appears on the blue balloon. At this point, the participant can either choose to keep pumping the balloon in order to increase the trial gains (by pressing the key ‘1’ the balloon becomes orange and can once again be pumped one to five times until it turns blue), or to cash-out the reward accumulated in the trial, permanently saving their earnings to a virtual bank account (by pressing the key ‘2’). In case they choose to cash-out their trial gains, the total amount saved so far on the virtual bank is displayed and a new trial begins (with an orange balloon).

The number of blue balloons for each trial was randomized between one and twelve, thus the probability of blue balloon exploding was defined as p=1/12-n with n= number of blue balloons. For example, the probability of exploding on the first blue balloon is 1/11, the second blue balloon is 1/10, and so on, until the 11th blue balloon. In order to encourage participants to rather keep pumping than cashing-out their earnings after the balloon turned blue, the reward doubled in case of a positive feedback, while only one point would be earned for each pump when the balloon was orange. Participants had the possibility of taking a small break between balloon blocks (every 20 balloon trials).”

Line 176. This is just a personal opinion, you might want to ignore. I noticed you used a 256 channels device to record neural data, but then you focus on just 6 of them. I think this is really a waste of information, especially when you will is to propose a “comprehensive” approach that should overcome the limitations of standard ones. In relation to this, you used a really retrograde and outdated analytic framework (e.g. ANOVA). You might be interested in more up-to-date approaches, such as whole-brain (although not necessarily) Multivariate Pattern Analysis.

We agree that using 256 channels only for source localization might appear as a waste of information. We could have used more complex approaches indeed, but we had to keep in mind that our goal was to detect neuromarkers for a future use in clinical practice. In that respect, standard EEG will be employed, so we kept standard analysis methods on a small set of electrodes. Moreover, such a classical approach made it possible to compare our results to those coming from other studies. We have nonetheless added this suggestion to our limitations section:

Lines 508-516: “Lastly, some limitations and strengths must be taken into consideration. This study is limited by a relatively small sample size, which could compromise generalizability of the findings, especially when it comes to the significant correlations observed. In addition, we lack of a model-based approach (such as considering cognitive parameters instead of summary measures, i.e. adjusted number of pumps) and more up-to-date analysis (such as whole-brain Multivariate Pattern Analysis) could have been applied in association with the BART and HR-EEG.  However, we chose a classical analytic approach to conduct this exploratory study in order to validate the neural markers of interest in our specific adaptation of the BART and have the possibility of establishing comparison with previous works in the same field, before reproducing the experiment in a larger trial involving patients suffering from Borderline personality disorder [77].”

Line 197. “The average number of blue balloons”. Do you mean the balloons exploded after pumping?

In our BART paradigm, only blue balloons can explode. Thus, the average number of blue balloons in our adaptation is the average number of blue balloons per trial over all trials that ended with the subject’s decision to cash-out (not by a balloon explosion).

We added more precision on lines 196-200: “In our adaptation of the BART program, the risk assessment was focused on the mean number of adjusted blue balloons only, because it is at this moment that participants have the possibility to place points in the virtual bank account or choose to keep pumping. Precisely, it corresponds to the average number of blue balloons per trial over all the trials that ended with the decision of cashing-out the accumulated points.”

Line 200. Honestly, I didn’t get when do you exactly pre-process EEG data. You did pre-processing on extracted epochs. If this is the case, why not pre-process the whole time series before epoching?

In fact, in Cartool we choose the filters at the time we extract epochs, but this is done on the whole signal. It is true that this appear confusing in the text. We modified the sentence by:

Lines 205-207: “Raw EEG data were re-referenced offline to a common average reference, a band pass filter was applied between 1 to 30 Hz and a notch filter was applied at 50 Hz to remove environmental artifacts.”

Line 230. Is there any particular reason why you filtered ERPs and not raw data? Maybe you wanted to refer to epochs, and not ERPs. Results change considerably if you decide to filter epochs or ERPs. You should discuss your choices more clearly

Thank you for the observation, we meant epochs indeed and not ERPs, but this sentence has been removed (see above).

Line 251. Is there any particular reason why you have chosen to perform one-way repeated measures ANOVA? Which is the variance between clusters (individuals). You might be interested in the Intra-Class correlation index, and based on that you should opt for a Mixed-Effects modeling approach.

We chose to perform repeated measures ANOVA for comparison purposes, because this method is classicaly used in other researches on the BART. Nevertheless, we understand that ICC can provide additional information, and we thank the reviewer for this idea. We measured  the Intra-Class correlation index, with two-way mixed effects, consistency, single rater/measurement.

ICC score was equal to 0.325 (95%-Confidence Interval: 0.091 < ICC < 0.61). This poor level of reliability might be interpreted has the consequence of the learning process occurring between blocks.

Therefore, in the methods section, we added the following sentences about ICC:

Lines 255-264: “The behavioral performance on our adaptation of he BART as it pertains to trials was taken into account by analyzing the task in four blocks with 20 trials each with a one-way repeat-ed-measures analysis of variance (ANOVA). The threshold of significance was set to 5% and post hoc analyses were performed using Bonferroni correction. In addition, we performed Inter-Class Correlation Coefficient (ICC) analyses using a two-way mixed effect model, with consistency and single rater per measurement. ICCs reflect the consistency of a measure taking into account variance related to the time of testing. Values below 0.50 are frequently considered to have a poor level of reliability, values from 0.50 to 0.75 to be of moderate reliability, and when higher than 0.75 they are considered to have very good reliability.”

In the results section, we added the following sentences:

Lines 278-281: “A block effect was detected on the average number of adjusted blue balloons (F(3.48)=9.51, p<0.00001). The first block’s average was significantly smaller in compari-son to the three following blocks (p<0.01 for all; see Figure 2). Moreover ICC score was equal to 0.325 (95%-Confidence Interval: 0.091 < ICC < 0.61), showing a poor level of reliability.”

Line 374. I think it is worth mentioning that the study lacks a model-based approach, which might be further investigated. Nowadays, we have sophisticated psychometric tools to model behavioral data, since extracting a summary measure (e.g. number of pumps) might not represent a valid account of some cognitive construct. One way to solve such a measurability issue is to use cognitive model decomposition to extract cognitive parameters from behavioral data. There are some examples using such an approach for the BART out there, such as, “D’Alessandro et al. (2020) – A joint modeling approach to analyze risky decisions by means of diffusion tensor imaging and behavioral data”; “Van Ravenzwaaij et al. (2011) – Cognitive model decomposition of the BART: assessment and application”, “Rolison et al. (2012) – Risky decision making in younger and older adults: the role of learning.”, to name just a few. The idea is to use cognitive parameters instead of summary measures for further analysis.

Thank you for this suggestion, we exect to apply this approach in our future studies. Nonetheless, we have added the suggestion in the limitations of our study:

Lines 523-527: “In addition, we lack of a model-based approach (e.g. considering cognitive parameters instead of summary measures, such as adjusted number of pumps), and more up-to-date analysis (e.g. whole-brain Multivariate Pattern Analysis) could have been applied in association with the BART and HR-EEG.”

Figures

Figure 4 and 6. You should report exactly what the grey areas refer to in the caption.

Thank you for this remark. We added the time windows for each ERP in the caption.

 “Figure 4. Grand average ERP waveforms after the onset of a loss and a win feedback on elec-trodes Fz and Cz. The FRN component appears between 275 and 330ms, while the P300 ERP oc-curs between 375 and 575ms after feedback.”    

“Figure 6. Influence of reward magnitude on the SPN and RewP components on the electrode Cz. a: Neural responses during the anticipation of smaller and larger rewards; a significantly more negative SPN is present for large rewards in the time window of -200 and 0ms before the feed-back appears. b: Neural response during reward processing; a significantly more positive RewP is observed following larger rewards, between 150 and 350ms after positive feedback.”

Typos and Formatting

Is there any particular reason (e.g. editorial) why you have reference indexes in isolated brackets, that is, “[i],[j]”, instead of “[i,j]”.

Why there is no spacing between a word and the reference index bracket, that is, “word[i]”, instead of “word [i]”.

Line 58. “an uniform” should be “a uniform”. In general, the English quality is poor for the whole sentence.

Line 124. Remove “+/-“ above “SD”, here and in other places.

Thank you for all the carefull suggestions, the formatting issues have been addressed.

Reviewer 3 Report

Summary: 

The current manuscript is aimed to analyze functional neuromarker of impulsivity. Results evidenced a positive correlation between the amplitude of the P300 component. This is an important topic and we need more research in this area in the future. However, some points need to be addressed.

Specific comments follow:

Introduction:

The introduction fits with the goal of the study. However, the state of the art is a bit poor and some other possible marker linked to decision making and impulsivity such as HRV should be reported

For example:

Forte, G., Morelli, M., & Casagrande, M. (2021). Heart Rate Variability and Decision-Making: Autonomic Responses in Making Decisions. Brain Sciences11(2), 243.

General comment: 

Generally, I found that  novelty and utility of the study were few emphasized. My advice to the authors is to highlight the innovative nature of the study. Furthermore, it would be interesting to provide some other implications of the results of this study.  

Author Response

Introduction:

The introduction fits with the goal of the study. However, the state of the art is a bit poor and some other possible marker linked to decision making and impulsivity such as HRV should be reported

For example: Forte, G., Morelli, M., & Casagrande, M. (2021). Heart Rate Variability and Decision-Making: Autonomic Responses in Making Decisions. Brain Sciences, 11(2), 243.

We thank you for this suggestion. We agree that there are indeed other possible markers linked to impulsivity that should be further explored and to be considered on our following research. However, as this study focused on the neurophysiology of risky decision-making with the application of ERP technique, we believe that including autonomic responses such as HRV would be out of this study’s scope.

General comment: Generally, I found that  novelty and utility of the study were few emphasized. My advice to the authors is to highlight the innovative nature of the study. Furthermore, it would be interesting to provide some other implications of the results of this study.

Thank you for this suggestion, we have modified the following passages accordingly:

Lines 543-553: “…we chose a classical analytic approach to conduct this exploratory study in order to vali-date the neural markers of interest specifically in our adaptation of the BART and have the possibility of establishing comparison with previous works in the same field, before re-producing the experiment in a larger trial involving patients suffering from Borderline personality disorder [78]. The study’s methodology is nonetheless strengthened by a rig-orous selection of non-clinical participants – which displayed varied impulsivity scores – and the application of validated assessment instruments. In addition, when compared to other adaptations of the BART, we believe that the presence of neutral (orange) balloons that must be pressed repeatedly represent an advantage given that it preserves the mo-tor-impulsivity component of the original version of the BART.”

Lines 566-580: “In this study, we demonstrated a direct relationship between ERP response in the context of risky decision-making and self-rated impulsivity. Our main finding concerned a positive correlation between a late feedback processing component (P300), in case of a positive feedback, and the UPPS global and (lack of) perseverance scores (Figure 5). Given the potential clinical application of these findings, further research is required towards a better understanding of both adaptive and maladaptive forms of risk-taking and impul-sive dimensions, allowing benefits on the approach and follow-up of impulsive behavior. Since a simple combination of different classes of the currently available instruments might not satisfactorily assess impulsivity in its multidimensional nature, the identifica-tion of precise dysfunctional mechanisms correlated to impulsive dimensions might con-tribute to the establishment of neuromarkers of impulsivity, as well as the development of diagnostic and therapeutic tools for impulsive-related disorders. Hence, trials investigat-ing ERP in the context of risk-taking, counting on larger sample sizes, assessing both clin-ical samples and controls, and applying model-based approaches, should be performed.”

Round 2

Reviewer 1 Report

Thank the authors for addressing my concerns.

1), regarding my comments on the self-report measures, can the authors add a brief description about what each subcomponent of the BIS-10 and UPPS measures? As a reader, I somehow understand what is motor-impulsivity, cognitive-impulsivity, etc., but am not sure whether my understanding is accurate or not. Can the authors introduce these as if the readers are not knowledgeable?

2), as my primary major concern, I commented that the associations between the behavioral results and the ERP should be strengthened, which might have been kind of confusing, sorry about that. What I meant was, compared to the current results reported in figures 4 & 6, the results on the association between the behavioral results of the task and the corresponding ERP may be more important, as least equally important compared to the correlation between UPPS and ERP. Given the ERP data are directly relevant to the neural basis of the task behaviors (or performance) and the authors' objective was developing neural markers, won't the association be critical?  

Author Response

Reviewer 1

Thank the authors for addressing my concerns.

1), regarding my comments on the self-report measures, can the authors add a brief description about what each subcomponent of the BIS-10 and UPPS measures? As a reader, I somehow understand what is motor-impulsivity, cognitive-impulsivity, etc., but am not sure whether my understanding is accurate or not. Can the authors introduce these as if the readers are not knowledgeable?

We sincerely thank the reviewers for the opportunity of improving our manuscript.

The manuscript lacked indeed of details on the subcomponents of the assessment scales. We have added the descriptions accordingly:

Lines 198-214: “From a trait-like perspective, impulsivity was measured through the BIS-10 [26]. The BIS-10 is a self-rated 34 item questionnaire, composed by three subscales: motor-impulsivity, cognitive-impulsivity, and non-planning-impulsivity. The subscale of motor-impulsivity evaluates actions without thinking, cognitive-impulsivity refers to making quick decisions, and non-planning-impulsivity assesses a lack of forethought (“futuring”). Each item is scored on a 0 to 4 points scale. Higher scores indicate higher levels of impulsivity. This study also included the UPPS [27], is a self-rated 45 item scale, evaluating the following dimensions: urgency, (lack of) premeditation, (lack of) perseverance, and sensation seeking. The urgency subscale assesses the tendency of the subject to feel strong impulsions, specially under negative affect; the (lack of) premeditation subscale evaluates the tendency to think about the consequences of an action before initiating it; the (lack of) perseverance, the capacity to sustain their attention during a task that may be hard or tedious; and the sensation seeking subscale involves two aspects: the tendency to enjoy and research exciting activities, and the opening to try new experiences that may or may not be dangerous. Each item is scored on a base of 4 points. The sum of all subscales (global UPPS score) is equally taken into account. Higher scores also indicate higher levels of impulsivity.”

2), as my primary major concern, I commented that the associations between the behavioral results and the ERP should be strengthened, which might have been kind of confusing, sorry about that. What I meant was, compared to the current results reported in figures 4 & 6, the results on the association between the behavioral results of the task and the corresponding ERP may be more important, as least equally important compared to the correlation between UPPS and ERP. Given the ERP data are directly relevant to the neural basis of the task behaviors (or performance) and the authors' objective was developing neural markers, won't the association be critical?

Thank you for this clarification, the authors agree with your questioning. A relationship between the behavioral component of the task and ERP data is indeed crucial, along with the relationship of ERP data with the applied self-report measures. We have detected an association between the risk-taking measure in the task (mean number of adjusted blue balloons) and the P300 response following a positive feedback on several adjacent electrodes with a p<.05, which was not treated as significant due to our rigorous significance criterion. However, we have modified our manuscript in order to strengthen the reliability of the ERP component (P300) as a potential neuromarker, encouraging further research, e.g. with larger samples sizes, that could confirm the significance of these simultaneous correlations:  

Figure 5.a was added: “Figure 5. Correlations with the P300 response (μV) after a positive feedback on the FPz elec-trode. a: Negative correlation with the mean number of adjusted blue balloons (r=-0.49, p<0.05). b: Significant positive correlation with the (lack of) perseverance subscale of the UPPS (r=0.79, p<0.001).”

We have equally adapted the following Discussion passage:

Lines 468-500: “An important result of the present study is that subjects who displayed smaller P300 amplitudes following gains scored less on the UPPS scale and on the lack of perseverance subscale, being thus less impulsive and having increased tendency to stay focused on a hard or tedious task, according to these measures. According to this reasoning, subjects who exhibited larger P300 amplitudes following gains – hence reducing the amplitude difference between their response to a loss and a gain – were more impulsive and lacked perseverance. In addition, it is worth mentioning that the P300 amplitude after a positive feedback displayed two correlations not considered significant in accordance with our criterion: (i) a negative correlation with the mean number of adjusted blue balloons (Figure 5.a), and (ii) a positive correlation with the UPPS lack of premeditation subscale (see Table 2), which suggests that these subjects could as well have less regard for the consequences of their actions. The described ERP results are consistent with our previously discussed behavioral data, which suggested that a greater mean number of adjusted blue balloons was curiously linked to increased premeditation ability (lower impulsivity scores), possi-bly indicating an adaptive risk-taking behavior. 

To the best of our knowledge, no other study has explored the correlations between the UPPS scale and ERP components in the BART. Nonetheless, previous studies apply-ing the BART had detected an association of a reduced difference between the P300 am-plitude following a negative and a positive feedback with impulsive-related behavior [41, 44]. This phenomenon was described in non-clinical samples: risky drivers vs safe drivers [44], and alcohol-intoxicated subjects vs controls [41]. However, similar results for the P300 were not observed on clinical subjects when it came to binge drinking [57] and alco-hol use disorder [49], in comparison to controls.

Considering the aforementioned observations, it is possible to hypothesize that changes on feedback-related responses could index impulsive-related behaviors. A reduc-tion on the amplitude difference between the P300 following a negative and a positive feedback could be linked to a spectrum of adaptive manifestations of impulsivity. Mala-daptive impulsive behavior, on the other hand, could be related to alterations on the FRN amplitude, as suggested by aforementioned studies. However, there is currently insuffi-cient data in order to draw solid conclusions, thus further research is needed, including larger sample sizes (clinical and non-clinical), which could confirm the significance of a simultaneous correlation of the P300 response with both behavioral and self-report measures.”

Reviewer 2 Report

The paper has improved significantly. I have just a few more comments.

Lines 304-306: You introduced the concept of inter-class correlation, which is not interesting in this case. I was asking for the intra-class correlation instead. The intra-class correlation indicates the amount of total variance in the data which is accounted by the between-cluster variance (clusters are individuals in your case). It simply indicates whether it might make sense, in a nested data structure (several individuals as clusters, and observations within each individual), to use multilevel modeling (mixed-effects modeling) and estimate explicitly the between-cluster variance. If the ICC is low, that means that clusters (individuals) are pretty homogeneous, and the bias in ignoring the multilevel structure of the data is reduced. Btw, if you can't handle the ICC (its computation or interpretation) remove it from the paper.

Lines 588-592: When you refer to model-based, you should highlight that it is related to the BART. Indeed, model-based analysis is not suitable for every behavioral task. But there are some chances to do model-based analysis with the BART since solutions exist in the literature. In general, or you elaborate a little bit more on the concept of "model-based", or you add citations. And please add also the citation for the MVPA (e.g. "Grootswagers et al. (2017) - Decoding Dynamic Brain Patterns from Evoked Responses: A Tutorial on Multivariate Pattern Analysis Applied to Time Series Neuroimaging Data"). Same for the uncertainty-risk disentangling paper.

Author Response

Reviewer 2

The paper has improved significantly. I have just a few more comments.

Lines 304-306: You introduced the concept of inter-class correlation, which is not interesting in this case. I was asking for the intra-class correlation instead. The intra-class correlation indicates the amount of total variance in the data which is accounted by the between-cluster variance (clusters are individuals in your case). It simply indicates whether it might make sense, in a nested data structure (several individuals as clusters, and observations within each individual), to use multilevel modeling (mixed-effects modeling) and estimate explicitly the between-cluster variance. If the ICC is low, that means that clusters (individuals) are pretty homogeneous, and the bias in ignoring the multilevel structure of the data is reduced. Btw, if you can't handle the ICC (its computation or interpretation) remove it from the paper.

We sincerely thank the reviewers for the opportunity of improving our manuscript.

We have unfortunately decided to remove the ICC data from the manuscript given the 3-days delay to reply.

Lines 588-592: When you refer to model-based, you should highlight that it is related to the BART. Indeed, model-based analysis is not suitable for every behavioral task. But there are some chances to do model-based analysis with the BART since solutions exist in the literature. In general, or you elaborate a little bit more on the concept of "model-based", or you add citations. And please add also the citation for the MVPA (e.g. "Grootswagers et al. (2017) - Decoding Dynamic Brain Patterns from Evoked Responses: A Tutorial on Multivariate Pattern Analysis Applied to Time Series Neuroimaging Data"). Same for the uncertainty-risk disentangling paper.

Thank you for the suggestion, we have added the references in the following passage:

Lines 551-556: “Lastly, some limitations and strengths must be taken into consideration. This study is limited by a relatively small sample size, which could compromise generalizability of the findings, especially when it comes to the significant correlations observed. In addition, we lack a model-based approach related to the BART (for examples, see [78–80]), and more up-to-date analysis (e.g. whole-brain Multivariate Pattern Analysis [81]) could be applied in association with HR-EEG.”

  1. van Ravenzwaaij, D., Dutilh, G. & Wagenmakers, E.-J. Cognitive model decomposition of the BART: Assessment and application. Journal of Mathematical Psychology 55, 94–105 (2011).
  2. Rolison, J. J., Hanoch, Y. & Wood, S. Risky decision making in younger and older adults: the role of learning. Psychol Aging 27, 129–140 (2012).
  3. D’Alessandro, M., Gallitto, G., Greco, A. & Lombardi, L. A Joint Modelling Approach to Analyze Risky Decisions by Means of Diffusion Tensor Imaging and Behavioural Data. Brain Sci 10, (2020).
  4. Grootswagers, T., Wardle, S. G. & Carlson, T. A. Decoding Dynamic Brain Patterns from Evoked Responses: A Tutorial on Multivariate Pattern Analysis Applied to Time Series Neu-roimaging Data. Journal of Cognitive Neuroscience 29, 677–697 (2017).

The paper by De Groot and Thurik (2018) has equally been added to our manuscript (see reference 13):

  1. De Groot, K. & Thurik, R. Disentangling Risk and Uncertainty: When Risk-Taking Measures Are Not About Risk. Front. Psychol. 9, (2018).